# The experience of trial participation disclosure among sex workers in a phase IIb HIV vaccine trial: A qualitative study in urban Tanzania

Joel Seme Ambikile[1]*, Edith A.M. Tarimo[2], Masunga K. Iseselo[1], Gift Lukumay[3], Patricia Munseri[4], Muhammad Bakari[4], Eligius Lyamuya[5], Said Aboud[5,6], Rachel Kawuma[7], Janet Seeley[7,8,9]

**1** Department of Clinical Nursing, Muhimbili University of Health and Allied Sciences, Dar es Salaam, Tanzania, **2** Department of Nursing Management, Muhimbili University of Health and Allied Sciences, Dar es Salaam, Tanzania, **3** Department of Community Health Nursing, Muhimbili University of Health and Allied Sciences, Dar es Salaam, Tanzania, **4** Department of Internal Medicine, Muhimbili University of Health and Allied Sciences, Dar es Salaam, Tanzania, **5** Department of Microbiology and Immunology, Muhimbili University of Health and Allied Sciences, Dar es Salaam, Tanzania, **6** National Institute for Medical Research, Dar es Salaam, Tanzania, **7** Medical Research Council/Uganda Virus Research Institute and London School of Hygiene and Tropical Medicine, Uganda Research Unit, social science, Entebbe, Uganda, **8** Africa Health Research Institute, KwaZulu-Natal, South Africa, **9** Faculty of Public Health and Policy, London School of Hygiene and Tropical Medicine, London, United Kingdom

☯ These authros are contributed equally.
\* joelambikile@yahoo.com

## Abstract

Globally, HIV vaccine clinical trials are conducted in the quest for an effective preventive vaccine. Volunteers' participation is vital to the success of these trials. However, disclosing involvement in a vaccine trial may have significant consequences, potentially affecting key aspects such as recruitment, retention, and overall engagement. This study aimed to explore the experiences of disclosure and non-disclosure of participation in a Phase IIb HIV Vaccine Trial among female sex workers in Dar es Salaam, Tanzania, and used a descriptive qualitative design. Fifteen in-depth interviews and four focus group discussions were conducted among volunteers who were participating in the HIV vaccine trial. Data analysis was done manually using the framework method. Three themes emerged: reasons for disclosure, reasons for non-disclosure, and consequences of disclosure. Reasons for disclosure were grouped into two categories: intended disclosure and unintended disclosure. Intended disclosure occurred to seek support for trial participation and to share information within trusted relationships. Unintended disclosure arose from circumstances related to trial participation. Reasons for non-disclosure had two categories: perceived lack of understanding about trial participation and concerns about inadequate support. Consequences of disclosure encompassed three categories: uncertainty about the vaccine's side effects, the perception of volunteers being infected with HIV, and disapproval of the vaccine trial. The findings reveal that volunteers experienced a

**Data availability statement:** All relevant data are within the manuscript and its Supporting Information files.

**Funding:** This study was funded by the second European and Developing Countries Clinical Trials Partnership (EDCTP2 to SA); (Grant reference: RIA2016 -1644). An additional funding support was also received from Gilead (CO-UK-412-5430 to SA). The funders had no role in study design, data collection and analysis, decision to publish, or preparation of the manuscript.

**Competing interests:** The authors have declared that no competing interests exist.

complex interplay between disclosure and non-disclosure of their participation in the PrEPVacc trial. The key themes – reasons for disclosure, non-disclosure, and consequences of disclosure – underscore the importance of understanding the personal and social factors influencing these decisions. These insights highlight the need for enhanced community education and support mechanisms to address concerns, mitigate misconceptions, and improve participation in HIV vaccine trials.

**Trial Registration:** This study was conducted as part of a multicenter phase IIb three-arm, two-stage HIV prophylactic vaccine trial with Registration Number NCT04066881, accessible at https://clinicaltrials.gov/study/NCT04066881.

# 1 Introduction

The 2022 statistics show that in Eastern and Southern Africa, 20.8 million people were living with HIV, 500,000 were newly infected, and 260,000 died from AIDS-related illnesses [1,2]. The 2023 statistics in Tanzania indicate that HIV prevalence is 4.4%, corresponding to approximately 1,548,000 adults living with HIV [3]. These statistics show that a significant burden of HIV and AIDS still exists and developing an effective HIV vaccines is a critical component in combating this epidemic [4].

Despite significant global advancements in addressing HIV and AIDS, including the crucial role played by antiretroviral therapy (ART), the disease remains incurable [5]. This underscores the ongoing search for an HIV preventive vaccine as a viable strategy to ultimately halt the pandemic [4]. Searching for an HIV vaccine continues to be one of the global health priorities, offering the potential for a definitive end to the HIV crisis. Numerous HIV vaccine efficacy trials conducted worldwide have provided promising insights, including on-going studies in the US [6–8], demonstrating that it is indeed possible to develop a safe and effective HIV vaccine. These trials have significantly contributed to the understanding of the intricate processes involved in vaccine development [9]. The successful conduct of these trials depends on the contributions of volunteer participants [10–12].

Researchers in Kenya observed that the volunteers' decision to participate in an HIV vaccine trial may be influenced by a complex interplay of personal and social factors, which can either facilitate or impede their involvement [10]. Other studies have shown that positive family and community attitudes towards participation serve as facilitators, while negative reactions from these social circles act as barriers. Concerns such as the potential repercussions of false HIV-positive results, uncertainties surrounding the trial process, side effects, the randomized assignment, trial duration, and uncertain efficacy are among the factors that may deter volunteers from participation [13–15]. Another study in Kenya showed that apprehensions can also elicit discouragement from family, friends, and community members, emphasizing the importance of volunteers carefully managing information about their participation to address misperceptions and assumptions surrounding vaccine research volunteers [16].

## 2 Materials and methods

### 2.1 Study design

This descriptive longitudinal qualitative study was conducted as part of a multicenter phase IIb three-arm, two-stage HIV prophylactic vaccine trial, with a secondary randomization performed to compare the efficacy of Tenofovir alafenamide/Emtricitabine (TAF/FTC), branded as descovy, with Tenofovir disoproxil fumarate/Emtricitabine (TDF/FTC), branded as Truvada, for use as pre-exposure prophylaxis (PrEP). Since the study investigated both PrEP and an HIV vaccine, it was referred to as the PrEPVacc trial. Our study is one of several qualitative studies conducted during the PrEPVacc trial. The COREQ checklist for reporting qualitative research interviews and focus groups was used in this study [17].

### 2.2 Study setting

The study took place in Dar es Salaam, the largest and fastest-growing commercial city in Tanzania, with a population of around 5.4 million [18,19]. Encompassing both urban and peri-urban areas, the Dar es Salaam city provided a diverse study population, including female sex workers. Specifically, data collection took place at Muhimbili University of Health and Allied Sciences (MUHAS), the premier and oldest public health university in Tanzania, whose core functions include training, research, and consultancy services. The university served as one of the sites for a multicenter phase IIb three-arm, two-stage HIV prophylactic vaccine trial.

### 2.3 Population

The study involved female sex workers who were enrolled in the PrEPVacc study in Dar es Salaam. Dar es Salaam City hosts 32% of the Tanzanian population, with a male-to-female ratio of 0.93 [19].

### 2.4 Sampling

The PrEPVacc Trial Qualitative Component Study Operations Manual (SOM) was used to guide the study sites to purposively sample 5–10% of the trial participants for the qualitative study based on characteristics such as age, sex, work type and PrEP regimen. The characteristics of the present study participants were age and PrEP regimen. The volunteers who met the criteria for in-depth interviews (IDIs) were selected through the assistance of a site Data Manager. The study's lead social scientist randomly selected 15 potential participants for interviews in the PrEPVacc trial at the Dar es Salaam-MUHAS site. This was done through weekly visits, during which one eligible volunteer was randomly selected using numbered slips of paper. The selected volunteer was then contacted to confirm her availability. If she was unavailable, the process was repeated, excluding her from the pool. Subsequently, counsellors were requested to arrange interview appointments with the selected participants. The participants were interviewed at three-time points during the trial – months 2, 6, and 12 – to explore their experiences with disclosure and non-disclosure of their participation in the Phase IIb HIV Vaccine Trial. The remaining volunteers who were not included in the IDIs, were purposefully invited to participate in focus group discussions (FGDs).

### 2.5 Data collection

Data collection for this study was done from 12th November 2021–5th May 2023 and involved 15 IDIs and four FGDs comprising 7–8 participants. Two experienced researchers, JSA and GL collected the data using the IDI and FGD guides. The guide(s) comprised of the following question(s): (1) Is there anyone who knows that you are participating in this study [PrEPVacc]? Who? (2) What do your partners/people around you think about your enrolment in the study? [what were the positive things that happened, challenges, concerns, and fears from partners/ people around] (3) How has your partner/

people around you supported your participation in the trial? All interviews and FGDs were conducted in Swahili and were audio-recorded. Field notes were also taken to capture nonverbal and environmental cues. Both IDIs and FGDs took place in a private and secured environment outside the trial clinic. Individual in-depth interviews (IDIs) lasted between 28 and 40 minutes, while FGDs lasted between 90 and 120 minutes. Data saturation was reached by the 10th IDI, although all 15 IDIs were completed.

## 2.6 Data management

Both IDIs and FGDs were conducted by experienced qualitative researchers. Prior to audio recording, written consent was obtained from all the volunteers for the use of recorders. Data were then securely transferred from the audio recorders to password-protected desktop computer at the study site. Each audio file was appropriately labeled and stored on the password-protected computer. The principal investigator cross-checked all transcripts against the audio-recorded information to ensure accuracy. The recordings were given pseudonyms before being sent to the transcribers. All audio recordings were transcribed verbatim by eight experienced transcribers. Transcripts were then anonymized and assigned unique identification numbers for each volunteer, and forwarded to an independent researcher for translation from Swahili to English. A team of four researchers familiar with the study reviewed the translated transcripts to ensure that the meaning was accurately conveyed and not distorted during translation. Both the Kiswahili and English transcripts were securely stored on password-protected computer. Hard copies of all forms, including consent forms, were securely stored under lock and key at the site.

## 2.7 Data analysis

We employed the framework method to analyze data [20]. The seven steps of qualitative data analysis described in this method were adopted as follows: (i) reading the transcripts iteratively to ensure deep immersion in the data (ii) familiarization with interviews by re-listening to the audio recordings (iii) coding through reading each transcript and writing labels to the data; (iv) developing a working analytical framework by compare the labels and agreeing on a set of codes to apply to all subsequent transcripts. by jotting down key concepts that were re-emerging from the data; (v) applying the analytical framework by indexing subsequent transcripts using the existing categories and codes; (vi) charting data into the framework matrix by summarizing the data by category from each transcript.; and (vii) interpreting the data. The first four researchers did data coding and engaged in discussions throughout the data analysis process, resolving any differences in thematic interpretation to reach a consensus. The original transcripts were frequently referenced during the consensus process to ensure accuracy and consistency in thematic interpretation. Given the time-consuming nature of this analysis method, meetings were held specifically for this purpose, particularly during the data interpretation stage.

## 2.8 Ethics considerations

The PrEPVacc study received approval from the National Health Research Ethics Committee (Approval number: NIMR/HQ/R.8a/Vol.IX/3333) and from the Muhimbili University of Health and Allied Sciences Research and Ethics Committee (Approval number: MUHAS-REC-11-2019-066) in Tanzania. Before participation in the qualitative study, the interviewer and potential participants reviewed the study information provided during previous visits, including what the qualitative study is about, to ensure participant understanding and recall. All potential study participants provided written consent to participate in the main HIV vaccine trial and its qualitative component. Additionally, they signed a separate consent form specifically for audio recording during the qualitative interviews. Participants were assured of their freedom to participate or withdraw from the study at any time, with the understanding that their decision would not impact the services they received during the study.

Global Public
Health

# 3 Results

## 3.1 Socio-demographic characteristics

A total of 15 IDIs and 4 FGDs, each comprising 7–8 participants, were conducted. All the study participants were females, aged between 20 and 37 years. The majority of IDIs participants were educated up to either two or four years of secondary education and single. Besides sex work, a few IDIs participants were engaging in other income-generating activities such as hairdressing, barmaid, petty business, and master of ceremony. The majority of FGDs participants in the two groups were educated for up to four years of secondary education, and the majority in the remaining two groups completed seven years of primary education.

## 3.2 Themes and categories

The volunteers' disclosure regarding their participation in the PrEPVacc trial unveiled three main themes: reasons for disclosure and nondisclosure, and consequences of disclosure. Reasons for disclosure were grouped into two categories: intended disclosure and unintended disclosure. Intended disclosure occurred to seek support for trial participation and to share information within trusted relationships. Unintended disclosure arose from circumstances related to trial participation. Reasons for non-disclosure had two categories: perceived lack of understanding about trial participation and concerns about inadequate support. Consequences of disclosure encompassed three categories: uncertainty about the vaccine's side effects, the perception of volunteers being infected with HIV, and disapproval of the vaccine trial. Each theme is elaborated upon in the subsequent paragraphs.

### 3.2.1 Reasons for disclosure. Intended disclosure

The volunteers revealed that that they disclosed their participation in the HIV vaccine trial to seek support for trial participation and to share information within trusted relationships.

***Seeking support for trial participation***

Volunteers' decision to disclose their participation in the PrEPVacc study were driven by various factors necessary to meet trial requirements. Disclosure was seen as important for enlisting support from people in their community, ensuring assistance in taking the pills, receiving timely reminders about trial visits, and fulfilling the trial's recommendation to share the consent form with family members.

To receive assistance in taking the pills, volunteers received support in storing the PrEP pills and in receiving timely reminders for medication, as one volunteer aptly expressed:

*Nearly one-third of people in my family know that I am participating in the study* [research]. *…My mother and father are diseased. But the relatives that I live with such as my sisters, brothers, and children of my relatives know that … so when it happens for example when I take the pill and leave it in the sitting room, they tell me that you have left your pills here, I tell them please keep them in the cupboard or the drawer. They keep them very well because they know what is going on* [in the trial]... (P2, IDI)

Disclosure was also crucial as it served as a means for volunteers to receive timely reminders about their trial visits from individuals in their living environment as verbalized by one volunteer:

*"My mum, brother and aunt support me in anything, sometimes when I forget the date of my attendance they remind me of that a day before my attendance"* (P7, IDI)

Some volunteers felt necessary to disclose their involvement in the trial due to their obligation to inform a parent or next of kin, coupled with the recommendation to share the consent form with family members. This disclosure was perceived as a mandatory requirement of the trial, as underscored below:

*Ah my uncle who is the young brother to my late mother...Yes, he does* [knowing about her participation in the trial]*, and remember that, when participating in the study there is a place where you are supposed to write the mobile phone numbers of people who are close to you and know about the study, so I wrote their mobile phone numbers and I should inform them whenever I come here* [trial site] *that I am going to Muhimbili [trial site]...* (P10, IDI)

### Sharing information within trusted relationships

The volunteers felt the need to disclose their trial participation, particularly to individuals they were close to, based on a foundation of trust. This circle of trusted individuals included sexual partners, parents, close relatives, friends, neighbors, fellow tenants, and other household acquaintances. The volunteers believed that the trustworthy relationships they shared warranted transparency, especially considering their familiarity with these close individuals as narrated in the following quote:

*My sister and one of my friends...I informed her about that, and also my friend; these are people that I told, that I am participating in the study...: I mean she is the child of my aunt; first of all, they are close people to me; secondly I trust them and they tell me about their issues I mean we are open about each other...if I am not available when they* [trial staff] *call me, they are the first ones to give me that information.* (P11, IDI)

Disclosure to close relatives, such as a mother, was similarly motivated by a volunteer's confidence in them as trustworthy individuals who were already knowledgeable about various aspects of the volunteer's life, including their involvement in sex work. This is exemplified by a volunteer who chose to disclose participation in the trial to her mother, whom she considered a confidant:

*Yes, at the end of the day she* [mother] *understood me after explaining to her [about participation in the trial], my mother has no problem. My mother understood me and she was the first one to encourage me. Because she is my confidant and knows everything about me, she knows even the kind of business* [sex work] *I am doing* (P14, IDI)

### Unintended disclosure

Some volunteers found themselves compelled to disclose their participation in the trial against their will due to various circumstances. These circumstances included the unintentional finding of the use of PrEP pills by others, the audible noise of the pill container, participating in the same trial with a close person, actively encouraging others to join the trial, the challenge of concealing pills from others, a desire to alleviate concerns from relatives, and the curiosity of others about frequent visits to the trial site. In some cases, the trial team's initial visits to gather volunteers' information inadvertently resulted in disclosing their participation. Many volunteers reported that the unintended detection of PrEP pill use emerged as one of the reasons for trial participation disclosure. This situation typically unfolded when close individuals, such as clients, relatives, or significant others, became aware of the pill consumption among the volunteers. In such instances, the volunteers felt compelled to inform these individuals about their participation in the PrEPVacc trial, as they had no other options besides addressing the situation transparently:

*My sister the one that I stay with saw the pills and at that time she was not in the study but now she is in this study; when she saw them, she said bad things about me, and asked me if those pills were for what? and the pills are in the tin like that and many other things. I brought her to the study site and they explained to her; she understood and later she joined the study.* (P7, IDI)

Some volunteers disclosed their participation in the trial due to the audible noise generated by the pill container. This occurred when volunteers had to carry the pills with them to fulfill the adherence per trial requirement, as explained by the following volunteer

"*I didn't disclose myself but because I am carrying them* [PrEP pills] *and they produce some sounds like 'chaka chaka chaka'; like ARVs, so they know.*" (P12, IDI)

Participation in the same trial with close relatives was reported as unavoidable situation for disclosure. Given their roles as study volunteers, they were required to engage each other during trial visits and meetings organized by the trial team. One volunteer, who had a sibling participating in the same trial, expressed the following:

*One of them* [siblings] *is participating in this study; she is the one who knows all my issues. She is the third born. I am with her in the study...So, I have told my mother about participating in the study, but my father knows nothing about it* (P15, IDI))

As part of their role in the PrEPVacc trial, the volunteers were involved in recruiting others for the study. Therefore, they felt obliged to disclose their participation in the trial to actively encourage others to join the study as reported below:

*There is one of my friends whom I motivated to get the vaccine. She asked me "You are motivating me to join, are you participating in it as well?" I told her I am also a volunteer and I asked her if she wanted to join I could take her there, that is the friend who knows that I am participating in this study* (P15, IDI)

Unintentional disclosure also occurred when volunteers faced difficulties concealing pills from others, ultimately leading to forced disclosure. The constant presence of other people made it difficult for a volunteer to discreetly take the pills. In such situations, a volunteer, who often spent time with friends, articulated the difficulty in hiding the act of taking the pills:

*So sometimes they* [friends] *come home and when we come back they sleep over. So, when the time comes they come to me so that we leave [for sex work] together. So before leaving for my business I had to take the pill that was it.... Yes, should I hide them [pills]? They see the pills while I take them... I told them that there is one study that is like this and that. The pills prevent HIV infection, but not by taking them one day, you are supposed to take them daily.* (P10, IDI)

The desire to alleviate concerns from their relatives also compelled volunteers to disclose their participation in the trial. The volunteers were mindful of potential worries from their family members regarding their involvement in the trial. Consequently, they found it necessary to disclose to minimize speculations, such as being perceived as unwell, as explained below:

*...I informed them* [close relatives] *about every stage of this study so that they should not be worried when I tell them I am going to Muhimbili. When you say you are going to Muhimbili its interpretation is that either you are sick or you are going to see a sick person, so I used to tell them that I am going to Muhimbili because there is this and that [issues of the trial].* (P2, IDI)

The curiosity of others about volunteers' frequent visits to Muhimbili triggered questions regarding where they were going, what they were doing there, and why. This compelled some volunteers to reveal the truth about the frequent visits and what they were exactly involved in as reported by the one whose mother was curious about her.

*I used to tell her that I am going somewhere, my mother I am in a certain place I will call you later. As a parent, she must have asked herself, "Why she has had many trips to Muhimbili, why is that?" That is it; she asked, "What do you do at Muhimbili?" I told her the truth that I am participating in a study which is looking for the HIV vaccine; as you might be aware our parents when you explain to them do understand a lot, so my mother has no problem....* (P14, IDI)

On the other hand, the trial team initially visited volunteers to gather information about them, including their location and household members. This led to relatives and others learning about the volunteers' participation in the trial, as reported by one of the participants in a focus group discussion:

*Some of my relatives know, I mean my grandmother, uncles, and my mother; know that because in the first phase when we were starting joining the study they used to come home. They used to come and see where you stay, who you live with, and how many people you live with. That is how my relatives knew about this…* (P7, FGD 3, poor adherers)

**3.2.2 Reasons for Non-disclosure.** The instances where volunteers chose not to disclose their participation in the trial were characterized by a perceived lack of understanding about trial participation and concerns about inadequate support.

**Perceived Lack of Understanding About Trial Participation**

Some volunteers refrained from disclosure because they believed that others, particularly close individuals such as sexual partners and parents, might not understand the nature of their participation in the trial. For some volunteers, the fear of being misunderstood stemmed from the suspicion that they would be thought to have been infected with HIV, leading to potential conflicts. One volunteer shared her concern that her boyfriend might not understand and that it could lead to arguments:

*To be honest, he* [the boyfriend] *doesn't know it* [participation in the trial]. *First of all, I think if you tell him he will not understand. He will not understand me, because let me give you a good example. Do you see this one, these patches on my skin, he saw them, I don't know what I ate, and he complained to me that I have HIV. Now, what will he do if he knows that I am participating in this? He saw the skin patches and told me I was HIV positive! And we quarrelled a lot because he told me that I am HIV positive, now what if he sees the pills?* (P11, IDI)

Another factor influencing volunteers' decision to withhold information about their trial participation was their status as parents. Some volunteers believed that their children were not in a position to comprehend the nature of their involvement. Additionally, there was concern that revealing the mother's participation in the trial might pose worries for the child regarding the mother's health, as expressed by one volunteer:

*The one who doesn't know is my child who is little, it is difficult to educate her about this when she sees me taking the drugs she asks Mum you are taking drugs, what are you suffering from?... so, because she is still a young child, I can't explain to her that I am participating in the study on HIV vaccine, PrEP etc. So, she has a challenge of not knowing what her mother is suffering from.* (P2, IDI)

**Concerns About Inadequate Support**

Nondisclosure was also influenced by concerns that disclosing their involvement might not result in the desired level of support from the informed ones. Some volunteers refrained from disclosure to avoid potential discouragement from close individuals, such as a sexual partner. This decision was illustrated by a volunteer who chose not to disclose her participation in the trial for a specific reason:

*He* [the partner] *discourages me. It doesn't matter how many times I tried to explain to him, he doesn't agree with it, he told me to go and have tests, but don't join the vaccine. It means that until now he doesn't know if I have joined the vaccine [study], he thinks that I just come here for normal attendance...* (P8, IDI)

Furthermore, the decision to withhold information was influenced by concerns that disclosing their involvement might lead to the exposure of their status as sex workers. Volunteers expressed apprehension about the potential

problems that might arise from the stigma associated with sex work and the reactions from others. One volunteer was afraid to inform her mother due to the perceived stigma and potential negative reactions associated with being a sex worker:

*Yeah, I don't think she* [the mother] *would understand it if I informed her, hahaha* [laughter]*.: I would have caused blood pressure* [hypertension] *to her. First of all, she would know that my daughter uses these pills because she is not settled; don't you know about parents! she would think that my daughter is a sex worker, I think she would not be happy about that.* (P11, IDI)

**3.2.3 Consequences of disclosure.** The act of disclosure had various consequences for the volunteers. These consequences were delineated by uncertainty about the vaccine's side effects, the perception of the volunteers being infected with HIV, and disapproval of the trial.

**Uncertainty About the Vaccine's Health Effects**

Disclosing participation in the trial led to uncertainty about the potential side effects of the vaccine, reflecting concerns and questions about its safety. Many people who learned about participation in the trial had worries and thoughts regarding the side effects of the vaccine and pills such as experiencing kidney and liver problems, infertility, cancer, and other unknown vaccine problems as put forward by one volunteer:

*For the first time, nobody understood* [participation in the trial] *because everyone was talking his/her own words, … some said I will be infertile, and some told me I will get cancer; I mean everyone was saying her/his own words …* (P12, IDI)

When some individuals learned about the volunteers' participation in the trial, they speculated to the extent of imagining peculiar outcomes, like volunteers transforming into chimpanzees or having their kidneys and livers taken away by the study team for undisclosed reasons, as claimed below:

*It happens sometimes, for example,* they [people who have learned about trial participation] *told me you will turn into a chimpanzee, sometimes I become worried, but I encouraged myself by saying there are others who have received the vaccine before me and they give testimonies in our meetings, so I get an encouragement of continuing to participate in the study.* (P12, IDI)

The ultimate concern for some individuals around the volunteers was the fear of potential loss of life. One volunteer reported that her mother, in particular, expressed a fervent hope that the vaccine she received would prove effective; otherwise, she would die.

*"I am glad of my relatives; they did not disagree [about participation in the trial]. Mostly they were telling me they wish that vaccine will work because they suspected they may lose me..."* (P1, IDI)

**Perception of Volunteers Being Infected with HIV**

There was a perception among individuals who knew about volunteers' trial participation that they might be HIV-positive, leading to worries about the outcomes. Many of these individuals believed that the volunteers contracted HIV through the vaccine, which they assumed the trial team had implanted in them, as articulated by one volunteer:

*"Some of my friends think that I have been infected with HIV because they say that they [researchers] inject you with HIV through those injections* (vaccines)." (P4, IDI)

The notion that volunteers might be deliberately infected with HIV by the trial team seemed to spread throughout the community. When some people learned about a volunteer's participation, they expressed confusion and suspected that the volunteers were being manipulated, as articulated by one participant in a focus group discussion:

*For me when I explained to my fellow workers that I was coming to this place because I am participating in the vaccine study at Muhimbili, they were puzzled and told me khaaa* [a term denoting being surprised]! *You are participating there. They will put the virus in the vaccine and then inject you, and then they will inject you another injection to remove the virus from you.* (P8, FGD 2, good adherers)

The circulation of rumours within the community about volunteers participating in the PrEPVacc trial being injected with HIV might have led to some opting to withdraw from the trial. Those who dropped out were reported to further propagate these rumours, asserting that volunteers were intentionally being infected with HIV, and these rumours aimed at discouraging others from participating in the trial. One volunteer said:

*They are our colleagues who were in this study, then they dropped out because of the rumours they were hearing, they heard that, we are being injected HIV and many carried the rumours which made them fail to continue participating in this study and dropped. Now they are spreading rumours to us so that we drop from the study as what they did.* (P11, IDI)

Thoughts of HIV infection emerged as volunteers used pills resembling those taken by individuals who are infected with HIV. Sexual partners, parents, friends, and neighbours, familiar with the appearance of antiretroviral drugs (ARVs), found it challenging to be convinced of the volunteers' HIV-negative status when they observed them using similar pills. One volunteer narrated:

*There is my neighbour, she knows that I am participating in this study; but the same neighbour says that you are HIV positive already; I tell her I am not HIV positive. She claims that there is another neighbour who says that she is using the same pills. So, these pills are the same as ARV, and if you say that you stop using, she asks you why don't you use them?* (P4, IDI)

The assumption that volunteers might have HIV stemmed from the belief that they joined the trial to receive medication for HIV treatment. This fuelled concerns that the volunteers' partners could become infected with HIV, thereby straining their relationships, as shared by a participant in a focus group discussion:

*I shared this with the father of my child that I am participating in a study in Muhimbili and the first thing that he thought directly was that this one is HIV positive already! So, a conflict arose and it was a big conflict in that situation we did not talk for about two days, he believed directly that I was HIV positive; he went to test for HIV and realized that he was HIV negative. … The problem was with this man, he thought directly that I was HIV positive, so, I joined the study to get preventive medicine against HIV infection.* (P3, FGD 3, poor adherers)

In some cases, volunteers encountered stigma and discrimination in the community because they were perceived as being HIV-positive, sometimes causing them to relocate to avoid these negative experiences, as mentioned by a participant in a focus group discussion:

*For me, it reached a stage where I was discriminated against, I was even given a room to sleep alone, I moved from home and now I am independent, and I am renting a room because it seemed that I will spoil my colleagues*

*because it reached a point that I go to refill my pills that meant I am HIV positive. So, I couldn't share water glass with other people, they tell you that no, buy your glass, spoon and plates so that you use them yourself.* (P4, FGD 6, good adherers)

**Disapproval of the Trial**

Disclosing participation also resulted in disapproval or skepticism about trial participation. The closely related people such as sexual partners, parents, siblings, and other relatives found it very difficult to accept the volunteers' participation in the trial.

*In the beginning, it was difficult for her* [sibling] *to accept [participation in the trial]. She used to quarrel a lot but as the days went on and the way I continued explaining to her, now she thinks that it is okay.* (P7, IDI)

The volunteers reported that some family members demonstrated disapproval of the trial by discouraging them from participation due to the adverse health outcomes they thought would be associated with using the trial products as articulated below:

*He [sexual partner] discourages me. It doesn't matter how many times I tried to explain to him, he doesn't agree with it, he told me to go and have tests, but not to join the vaccine…my mother refused and said they are infecting you with the virus and then they will inject you that injection I explained to her and she understood … I explained to him [father] and understood although he didn't agree quickly... they told me don't participate in the study because they will give you an injection, do you see these pills which prevent HIV?* (P8, IDI)

## 4 Discussion

The study aimed to investigate the dynamics of the disclosure of volunteers' participation in a Phase IIb HIV Vaccine Trial. The findings highlight both intended and unintended reasons for disclosure, including seeking support for trial participation and sharing information within trusted relationships. On the other hand, non-disclosure was driven by a perceived lack of understanding about the vaccine trial and concerns over inadequate support. The consequences of disclosure included uncertainty about the vaccine's health effects, perceptions of volunteers being HIV positive, and disapproval of the trial. These insights emphasize the need for future strategies, including community education, to enhance volunteers' participation in HIV and other vaccine trials.

Seeking support for trial participation was one of the primary reasons for volunteers' intended disclose to their participation in the HIV vaccine trial. This disclosure was considered essential for securing assistance in taking the pills, receiving timely reminders about trial visits, and fulfilling the trial's recommendation to share the consent form with family members. These factors highlight the critical role that social support plays in participants' ability to successfully adhere to the trial requirements, indicating that disclosure is not merely a personal decision but as a necessary step to ensure compliance with trial protocols. However, a previous study has shown that obtaining such support from family and friends can be challenging, especially when families disagree with the volunteer's participation, potentially leading to strained relationships within the family and community [21]. Research on vaccine trial participation has further affirmed that social networks, including friends, family, colleagues, and communities, can significantly influence participation decisions, either positively or negatively [22]. Additionally, our findings also suggest that trial protocols could impact volunteers' decisions regarding disclosure. For instance, a study conducted in Uganda indicated that volunteers felt compelled to reveal their involvement in an HIV clinical trial due to the informed consent process, which involved discussions among family members [23]. This underscores the ethical considerations surrounding disclosure, as volunteers may feel pressured to share their participation, which could affect their privacy and autonomy [24–26]. Therefore, disclosing trial participation should be approached

with caution to mitigate potential undesired consequences, such as ethical concerns and a lack of support from close individuals.

Another primary reason for volunteers' intended disclosure to their participation in the HIV vaccine trial was sharing information within trusted relationships. They felt an obligation to disclose their trial participation to close individuals, including partners, parents, relatives, friends, neighbors, fellow tenants, and other household acquaintances. This highlights the significance volunteers placed on crucial social connections and networks, emphasizing the need for transparency. Sharing such sensitive information becomes crucial to preempt potential conflicts stemming from a lack of information. In Uganda, for instance, male volunteers informed their partners about trial participation due to the necessity of attending follow-up visits at the clinic, aiming to avoid subsequent conflicts [23]. A similar sentiment was echoed in a study in India, where men who have sex with men (MSM) emphasized the importance of informing family members about their participation in an HIV vaccine trial to secure approval. This proactive communication was seen as a means to prevent later discord within the family, especially if volunteers encountered medical complications during the trial [27].

There are circumstances that led to unintended trial participation disclosure in our study including inadvertent discovery of PrEP pill usage by others, the audible noise emanating from the pill container, and the shared participation in the same trial with a close individual. These highlight considerable challenges volunteers face in maintaining complete confidentiality regarding their participation in a vaccine trial. This resonates with findings from a previous study where unintended disclosure occurred due to others spotting the volunteer at the clinic, family members stumbling upon study materials at home, and the discovery of a volunteer's vaccine study identification card [28]. Unintended disclosure of trial participation appears to be common among volunteers in vaccine trials. This phenomenon can lead to negative experiences for volunteers, including challenges in interpersonal relationships, negative comments, and instances of stigmatization or discrimination [29,30]. Addressing these challenges is crucial to ensure a supportive and respectful environment for trial volunteers.

Volunteers cited various reasons for choosing not to disclose their participation in the vaccine trial. Some expressed concerns about being misunderstood by their partners and parents, fearing that the revelation might lead to unfounded assumptions of HIV infection and subsequent conflicts. Others believed that their children were not capable of grasping the nature of their involvement. This highlights the precautionary measures volunteers took to navigate their participation in the vaccine trial without interference from their partners or close relatives. A parallel concern was noted in a previous HIV vaccine trial among adolescents in the U.S., where participants were hesitant to inform their potential partners due to skepticism about their understanding of the trial's implications. This suggests a common thread of apprehension surrounding disclosure in vaccine trials, particularly when it comes to ensuring comprehension and avoiding potential conflicts [31].

Another reason for nondisclosure was the desire to maintain support from close individuals, as volunteers feared that disclosure might disrupt this support and potentially lead to discouragement. This underscores the substantial importance volunteers placed on the support they received from those close to them throughout the trial. In the beginning of the same HIV vaccine trial, it was observed that some volunteers did not attend subsequent sessions, the major reason being advice from their partners or family members [12], highlighting the considerable impact of significant others.

The expressed concerns that disclosing volunteers' involvement in the trial could lead to the exposure of their status as sex workers, resulting in stigma and negative reactions from others is worth noting. This highlights the volunteers' determination to shield themselves from potential social harm that could arise from revealing their occupation as sex workers. This may include avoidance of legal implications of their job which is illegal in Tanzania. Similar concerns have been noted in other HIV vaccine trials, particularly among MSM. In these instances, participants were worried about their status becoming known to others, including parents and insurance companies, which could result in the denial of certain services such as health insurance. The fear of being identified as gay, misconstrued as having HIV, and being judged as "promiscuous" contributed to their reluctance to disclose their participation in the trials [13,27].

An important revelation from the present study was the aftermath of disclosure, with one notable consequence being the concern of volunteers' close individuals regarding the potential health effects of the vaccine. This underscores their apprehension about the safety of the volunteers' health, expressing worries that the administered vaccine and pills could lead to kidney and liver problems, infertility, cancer, and other unforeseen side effects. Similar concerns among the close individuals of volunteers have been reported in comparable studies conducted in Tanzania, Kenya, Guinea, as well as in systematic reviews. These studies have consistently highlighted concerns related to the fear of unknown side effects of vaccines, overall distrust, and the potential for social discrimination [11,16,32–34]. In some instances, documented in these studies, volunteers declined participation later due to the fear associated with these unknown side effects. This emphasizes the imperative to educate the community and address these concerns to foster greater participation in vaccine trials.

Another consequence of trial participation disclosure was a prevailing perception among individuals to whom volunteers disclosed their trial participation that the volunteers might be HIV-infected, leading to anxieties about potential negative outcomes. Many of these individuals believed that the volunteers had contracted HIV through the vaccine, suspecting intentional plans by the trial team to implant HIV to them. This highlights a significant gap in education and widespread misconceptions within the community regarding vaccine trials. Similar findings have been reported in various studies conducted in other countries such as Kenya, China, and the U.S. In these studies, family and community members have asserted that the vaccine itself caused the volunteer to become HIV-infected or more susceptible to infection [16,28,31,35]. Again, this calls for the need to educate the community regarding vaccine trials to avoid such beliefs which may discourage volunteers' participation.

A further consequence of trial participation disclosure was the disapproval or skepticism expressed by volunteers' close associates, including sexual partners, parents, siblings, and other relatives. The fear of unknown side effects, coupled with concerns that volunteers might have been intentionally infected with HIV as previously described, likely contributed to these reactions. Similar findings have been documented in previous studies. Volunteers in these studies reported facing resistance from significant others such as fiancées, partners, parents, relatives, colleagues, and friends due to worries about potential side effects and misunderstandings regarding the participant's HIV status or risk of infection [29,33]. This underscores the importance of community educational interventions on vaccine trials to address and mitigate the resistance from individuals who may discourage volunteers' participation.

## 5 Conclusion

The findings of this study reveal that volunteers navigated a complex interplay between disclosure and non-disclosure of their participation in the PrEPVacc trial. Social bonds and support systems, trial requirements, and unavoidable trial-related circumstances played pivotal roles in influencing decisions around disclosure. These insights emphasize the need for tailored community education programs that address misconceptions about trial participation, and provide clear guidance on how to manage disclosure. Furthermore, support mechanisms should be developed to assist participants in navigating family and community dynamics, reducing the pressure to disclose, and protecting their autonomy and privacy. Such mechanisms could include counseling services, community sensitization activities, and structured peer-support networks to help participants manage social expectations and relational tensions arising from trial involvement. Strengthening these resources can improve engagement and retention in HIV vaccine trials while fostering a more supportive environment for volunteers.

## Supporting information

**S1 File. COREQ checklist.**
(PDF)

## Author contributions

**Conceptualization:** Joel Seme Ambikile, Edith A.M. Tarimo, Masunga K. Iseselo, Gift Lukumay.

**Data curation:** Joel Seme Ambikile, Edith A.M. Tarimo, Masunga K. Iseselo, Gift Lukumay.

**Formal analysis:** Joel Seme Ambikile, Edith A.M. Tarimo, Masunga K. Iseselo, Gift Lukumay.

**Funding acquisition:** Said Aboud.

**Investigation:** Joel Seme Ambikile, Edith A.M. Tarimo, Gift Lukumay.

**Methodology:** Joel Seme Ambikile, Edith A.M. Tarimo, Masunga K. Iseselo, Gift Lukumay.

**Project administration:** Patricia Munseri, Muhammad Bakari, Said Aboud.

**Resources:** Edith A.M. Tarimo.

**Supervision:** Edith A.M. Tarimo, Muhammad Bakari, Said Aboud, Rachel Kawuma.

**Validation:** Rachel Kawuma, Janet Seeley.

**Writing – original draft:** Joel Seme Ambikile.

**Writing – review & editing:** Joel Seme Ambikile, Edith A.M. Tarimo, Masunga K. Iseselo, Gift Lukumay, Patricia Munseri, Muhammad Bakari, Eligius Lyamuya, Said Aboud, Rachel Kawuma, Janet Seeley.

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
