## [Decision Letter · Decision Letter 0]

25 Jun 2025

PGPH-D-25-00311

The Experience of Trial Participation Disclosure Among Sex Workers in a Phase IIb HIV Vaccine Trial: A Qualitative Study in Urban Tanzania

Dear Dr. Ambikile,

Thank you for submitting your manuscript to PLOS Global Public Health. After careful consideration, we feel that it has merit but does not fully meet PLOS Global Public Health’s publication criteria as it currently stands. Therefore, we invite you to submit a revised version of the manuscript that addresses the points raised during the review process.

Please note that we have only been able to secure a single reviewer to assess your manuscript. We are issuing a decision on your manuscript at this point to prevent further delays in the evaluation of your manuscript. Please be aware that the editor who handles your revised manuscript might find it necessary to invite additional reviewers to assess this work once the revised manuscript is submitted. However, we will aim to proceed on the basis of this single review if possible. 

Please carefully consider the reviewer's comments and revise your manuscript as appropriate, providing a point-by-point response to their suggestions upon resubmission. Please do carefully address the question about sampling and randomization, providing additional details about the process.

We look forward to receiving your revised manuscript.

Kind regards,

Sarah Jose, Ph.D.

Staff Editor

Journal Requirements:

1. Please ensure you have included the registration number for the clinical trial referenced in the manuscript.

2. Your current Financial Disclosure states, “This study was funded by The Second European & Developing Countries Clinical Trials Partnership (EDCTP2); (Grant reference: RIA2016 -1644). An additional funding support was also received from Gilead (CO-UK-412-5430).”. However, your funding information on the submission form indicates that you received funding from “Center for Clinical Trials, Japan Medical Association”. Please indicate by return email the full and correct funding information for your study and confirm the order in which funding contributions should appear. Please be sure to indicate whether the funders played any role in the study design, data collection and analysis, decision to publish, or preparation of the manuscript.

3. Your manuscript is missing the following sections: Result. Please ensure these are present, and in the correct order, and that any references to subheadings in your main text are correct. An outline of the required sections can be consulted in our submission guidelines here: 

https://journals.plos.org/globalpublichealth/s/submission-guidelines#loc-parts-of-a-submission

Reviewers' comments:

Reviewer's Responses to Questions

**Comments to the Author**

1. Does this manuscript meet PLOS Global Public Health’s publication criteria?

Reviewer #1: Yes

2. Has the statistical analysis been performed appropriately and rigorously?

Reviewer #1: Yes

3. Have the authors made all data underlying the findings in their manuscript fully available (please refer to the Data Availability Statement at the start of the manuscript PDF file)?

Reviewer #1: Yes

4. Is the manuscript presented in an intelligible fashion and written in standard English?

Reviewer #1: Yes

Reviewer #1: This is an excellent paper on the reasons for disclosing/not disclosing HIV vaccine participation among sex workers, and I feel that it may be a valuable contribution to the body of literature on stigma surrounding HIV and its treatment, especially among special populations such as the one focused on in this study.

I was especially impressed by the readability of the data shared - there are a variety of testimonies included, and the categorization of the different reasons and flow of the narrative was easy to read. The conclusions were sound, and the implications for further research into HIV vaccines was thought-provoking.

My sole minimal concern is that it may be beneficial to include a description of the randomization process, especially as the sample size is relatively small

All in all, the submitted manuscript reads very well, and I have no major concerns about the quality of the paper.

**Do you want your identity to be public for this peer review?** For information about this choice, including consent withdrawal, please see our Privacy Policy

Reviewer #1: **Yes: ** Abigail Jeyaraj

---

## [Decision Letter · Decision Letter 1]

7 Sep 2025

PGPH-D-25-00311R1

The Experience of Trial Participation Disclosure Among Sex Workers in a Phase IIb HIV Vaccine Trial: A Qualitative Study in Urban Tanzania

Dear Dr. Ambikile,

Thank you for submitting your manuscript to PLOS Global Public Health. After careful consideration, we feel that it has merit but does not fully meet PLOS Global Public Health’s publication criteria as it currently stands. Therefore, we invite you to submit a revised version of the manuscript that addresses the points raised during the review process.

The manuscript has been evaluated by two reviewers, and their comments are available below. The reviewers have raised a number of concerns. Could you please carefully review the comments and revise the manuscript to address all comments raised?

We note that one of the reviewers requested citation. We recommend that you please review and evaluate the requested works to determine whether they are relevant. It is not a requirement to cite these works

We look forward to receiving your revised manuscript.

Kind regards,

Katrien G. Janin, PhD

Staff Editor

Journal Requirements:

Reviewers' comments:

Reviewer's Responses to Questions

**Comments to the Author**

Reviewer #1: All comments have been addressed

Reviewer #2: All comments have been addressed

publication criteria?

Reviewer #1: Yes

Reviewer #2: No

3. Has the statistical analysis been performed appropriately and rigorously?

Reviewer #1: Yes

Reviewer #2: No

4. Have the authors made all data underlying the findings in their manuscript fully available (please refer to the Data Availability Statement at the start of the manuscript PDF file)?

Reviewer #1: Yes

Reviewer #2: No

5. Is the manuscript presented in an intelligible fashion and written in standard English?

Reviewer #1: Yes

Reviewer #2: Yes

Reviewer #1: (No Response)

Reviewer #2: The review gives a lot of background but doesn’t clearly highlight what new perspective this review adds. Right now, it feels like a collection of summaries. You should state early (in the introduction) what unique lens you’re applying compared to other reviews. Please refine and enrich your introduction by taking help from this paper and citing it. https://doi.org/10.1007/s10142-025-01613-1. https://doi.org/10.1111/cbdd.70121. https://doi.org/10.1111/cbdd.70121. doi.org/10.1007/s10989-025-10702-5. https://doi.org/10.1371/journal.pone.0317382. https://doi.org/10.1002/cpdd.1585.doi.org/10.1007/s10142-024-01425-9

Some points (gut barrier damage, microbial metabolites, dysbiosis, SCFAs) are repeated in multiple sections almost word-for-word. This makes the paper feel longer than it needs to be. These could be consolidated to make the manuscript sharper and easier to read

The bacteria section is very detailed, but fungi and viruses get less critical attention. The imbalance makes those parts feel incomplete. Either expand with more critical evaluation or shorten them so expectations are aligned.

Many sections describe results (“X study found this, Y study found that”), but there’s little critical discussion. For example: when probiotics show “mixed results,” the paper doesn’t explain why. A review should analyze possible reasons (differences in study design, ART status, geography, diet, etc.).

The tables are useful but very text-heavy. Instead of repeating sentences, simplify to key points (“enriched,” “depleted”). This will make them faster to read.

Figure 2 (gut–liver axis) is interesting but doesn’t clearly connect to HIV. Without a stronger link, it feels like a side topic.

Abbreviations should be consistent (e.g., CD4⁺ vs CD4+ T cells, ART vs antiretroviral therapy).

Some superscripts and subscripts are inconsistent. Standardize them.

Some references are older (2014–2017). Updating with more recent studies would make the review stronger.

A few sources are cited multiple times for similar points; consider diversifying references.

**Do you want your identity to be public for this peer review?** For information about this choice, including consent withdrawal, please see our Privacy Policy

Reviewer #1: **Yes: ** Abigail Jeyaraj

Reviewer #2: No

---

## [Decision Letter · Decision Letter 2]

21 Oct 2025

PGPH-D-25-00311R2

The Experience of Trial Participation Disclosure Among Sex Workers in a Phase IIb HIV Vaccine Trial: A Qualitative Study in Urban Tanzania

Dear Dr. Ambikile,

Thank you for submitting your manuscript to PLOS Global Public Health. After careful consideration, we feel that it has merit but does not fully meet PLOS Global Public Health’s publication criteria as it currently stands. Therefore, we invite you to submit a revised version of the manuscript that addresses the points raised during the review process.

Please review and respond to the comments below (editor, and second reviewer).

We look forward to receiving your revised manuscript.

Kind regards,

Stephen Bell

Academic Editor

Journal Requirements:

Please ensure you have included the registration number for the clinical trial referenced in the manuscript.

Additional Editor Comments (if provided):

Please see attached comments from the second reviewer and make any minor revisions accordingly.

Please also integrate a recognised check list for reporting qualitative studies. PLOS Global Public Health considers qualitative and mixed-methods studies for publication. We recommend that authors use the COREQ checklist, or other relevant checklists listed by the Equator Network, such as the SRQR, to ensure complete reporting (http://journals.plos.org/globalpublichealth/s/submission-guidelines#loc-qualitative-research). In general, we would expect qualitative studies to include the following: 1) defined objectives or research questions; 2) description of the sampling strategy, including rationale for the recruitment method, participant inclusion/exclusion criteria and the number of participants recruited; 3) detailed reporting of the data collection procedures; 4) data analysis procedures described in sufficient detail to enable replication; 5) a discussion of potential sources of bias; and 6) a discussion of limitations.

Reviewers' comments:

Reviewer's Responses to Questions

**Comments to the Author**

Reviewer #1: All comments have been addressed

Reviewer #3: (No Response)

publication criteria?

Reviewer #1: (No Response)

Reviewer #3: Yes

3. Has the statistical analysis been performed appropriately and rigorously?

Reviewer #1: (No Response)

Reviewer #3: N/A

4. Have the authors made all data underlying the findings in their manuscript fully available (please refer to the Data Availability Statement at the start of the manuscript PDF file)?

Reviewer #1: (No Response)

Reviewer #3: Yes

5. Is the manuscript presented in an intelligible fashion and written in standard English?

Reviewer #1: (No Response)

Reviewer #3: Yes

Reviewer #1: (No Response)

Reviewer #3: (No Response)

**Do you want your identity to be public for this peer review?** For information about this choice, including consent withdrawal, please see our Privacy Policy

Reviewer #1: **Yes: ** Abigail Jeyaraj

Reviewer #3: No

---

## [Editor Report · Decision Letter 3]

3 Nov 2025

PGPH-D-25-00311R3

The Experience of Trial Participation Disclosure Among Sex Workers in a Phase IIb HIV Vaccine Trial: A Qualitative Study in Urban Tanzania

Dear Dr. Ambikile,

Thank you for submitting your manuscript to PLOS Global Public Health. After careful consideration, we feel that the reviewers' comments have been adequaetly dealt with, but a response to the editor feedback is missing. Therefore, we invite you to submit a revised version of the manuscript that addresses the prior points raised during the review process.

EDITOR: Thank you for responding to the reviewers' comments. This is nearly there, but there was no revision made based on the editorial feedback. We have added this in here once more. Please can you resolve this by (a) making edits in the paper in line with this feedback (i.e. stating that a checklist like COREQ or SRQR was used), and (b) providing a supplementary document (i.e the checklist table; you will find examples of this in other qualitative publications in this journal) that can be added to the publication when it is published:

Please also integrate a recognised check list for reporting qualitative studies. PLOS Global Public Health considers qualitative and mixed-methods studies for publication. We recommend that authors use the COREQ checklist, or other relevant checklists listed by the Equator Network, such as the SRQR, to ensure complete reporting (http://journals.plos.org/globalpublichealth/s/submission-guidelines#loc-qualitative-research). In general, we would expect qualitative studies to include the following: 1) defined objectives or research questions; 2) description of the sampling strategy, including rationale for the recruitment method, participant inclusion/exclusion criteria and the number of participants recruited; 3) detailed reporting of the data collection procedures; 4) data analysis procedures described in sufficient detail to enable replication; 5) a discussion of potential sources of bias; and 6) a discussion of limitations.

A rebuttal letter that responds to each point raised by the editor. You should upload this letter as a separate file labeled 'Response to Reviewers'.A marked-up copy of your manuscript that highlights changes made to the original version. You should upload this as a separate file labeled 'Revised Manuscript with Track Changes'.An unmarked version of your revised paper without tracked changes. You should upload this as a separate file labeled 'Manuscript'.The supplementary document

We look forward to receiving your revised manuscript.

Kind regards,

Stephen Bell

Academic Editor
---

## [Editor Report · Decision Letter 4]

6 Nov 2025

The Experience of Trial Participation Disclosure Among Sex Workers in a Phase IIb HIV Vaccine Trial: A Qualitative Study in Urban Tanzania

PGPH-D-25-00311R4

Dear Dr. Ambikile,

We are pleased to inform you that your manuscript 'The Experience of Trial Participation Disclosure Among Sex Workers in a Phase IIb HIV Vaccine Trial: A Qualitative Study in Urban Tanzania' has been provisionally accepted for publication in PLOS Global Public Health.

Best regards,

Stephen Bell

Academic Editor